# Development of 3D-Bioprinted Colitis-Mimicking Model to Assess Epithelial Barrier Function Using Albumin Nano-Encapsulated Anti-Inflammatory Drugs

**DOI:** 10.3390/biomimetics8010041

**Published:** 2023-01-18

**Authors:** Abdulmajeed G. Almutary, Abdullah M. Alnuqaydan, Saleh A. Almatroodi, Hamid A. Bakshi, Dinesh Kumar Chellappan, Murtaza M. Tambuwala

**Affiliations:** 1Department of Medical Biotechnology, College of Applied Medical Sciences, Qassim University, Buraydah 51452, Saudi Arabia; ami.alnuqaydan@qu.edu.sa; 2Department of Medical Laboratories, College of Applied Medical Sciences, Qassim University, Buraydah 51452, Saudi Arabia; smtrody@qu.edu.sa; 3The Hormel Institute, Medical Research Center, University of Minnesota, Austin, MN 55912, USA; bakshi-h@ulster.ac.uk; 4Department of Life Sciences, School of Pharmacy, International Medical University, Kuala Lumpur 57000, Malaysia; dinesh_kumar@imu.edu.my; 5Lincoln Medical School, University of Lincoln, Brayford Pool Campus, Lincoln LN6 7TS, UK

**Keywords:** 3D bioprinting, colitis 3D model, drug prediction, barrier function, albumin nanoparticles, Roxadustat, CAPE

## Abstract

Physiological barrier function is very difficult to replicate in vitro. This situation leads to poor prediction of candidate drugs in the drug development process due to the lack of preclinical modelling for intestinal function. By using 3D bioprinting, we generated a colitis-like condition model that can evaluate the barrier function of albumin nanoencapsulated anti-inflammatory drugs. Histological characterization demonstrated the manifestation of the disease in 3D-bioprinted Caco-2 and HT-29 constructs. A comparison of proliferation rates in 2D monolayer and 3D-bioprinted models was also carried out. This model is compatible with currently available preclinical assays and can be implemented as an effective tool for efficacy and toxicity prediction in drug development.

## 1. Introduction

In the drug development process, systemic availability, off-target outcomes, and reduced efficacy comprise the main challenges for the successful clinical prediction of candidate drugs [1]. This situation highlights the need for superior preclinical tools so that intestinal function can be modelled [2]. For drug administration, oral delivery is the most common route, and in addition to absorption and metabolism, an appropriate location for specific substances, namely, nonsteroidal anti-inflammatory drugs (NSAIDs) and other therapeutic compounds, is directly provided by the intestine [3,4]. It also functions as an organ for studying interactions between drugs [5,6]. Animal models are often used to determine the toxicity and efficacy of drugs. However, species differences lead to a disparity in translating results to clinical settings [7].

The major in vitro models employed to investigate intestinal bioavailability and toxicity are 2D monolayers and microsomes [8]. Although microsomes serve as a viable model for the initial assessment of metabolism, they are not able to model cellular-level outcomes [9]. In addition, the lack of physiological conditions in the 2D model leads to poor clinical predictions [10]. Newer models using microfluidic chips enable enhancement of several physiological conditions that appear to be lacking in 2D cultures [11]. Despite this, cell-based studies and molecular signalling experiments still remain a challenge [12]. In vitro models have certain limitations that have affected the refinement of techniques involving human primary cell lines of intestinal origin [13]. In contrast to cell lines, monocultures resemble actual tissues. Isolation of the epithelium from supportive cells can do two things: first, it may impair functioning; second, it may restrict the ability to have complex cell interactions, including inflammatory responses or interstitial cells [14].

When organoids were discovered, this revealed another path to modelling the intestine in vivo [15]. However, they may lack organ physiology in vivo due to an inward epithelium, making them inaccessible for pharmacokinetic assays such as absorption, distribution, metabolism, and excretion [16]. For this reason, the creation of accessible compartments requires a complex bioreactor setup and producing a monolayer with nonphysiological transepithelial electric resistance (TEER) [17]. To overcome these limitations that are evident in the existing system, a commercial bioprinting source with appropriate technology was developed for a complex three-dimensional human organ model so that it may effectively imitate the native structure and function of the organ [18]. In this study, we attempt to develop a 3D-bioprinted model for colitis to evaluate the efficacy of anti-inflammatory drugs on barrier function in 2D- and 3D-bioprinted models. Cell growth in 2D monolayers allows access to the same amount of nutrients and growth factors present in the medium, leading to homogeneous growth and cellular proliferation. The key novelty of this model is its simplicity to use and efficiency. Because the major limitation of 2D cell culture methods for mimicking colitis is that they do not have any control over shape and structure, TEER measurement cannot be accurate for measuring barrier function. In our model, the control of the shapes of cells, adhesive islands, microwells, and micropillars has been customized via a 3D bioprinting method that will allow for reproducible results in terms of measurement of the epithelial barrier function.

## 2. Materials and Methods

### 2.1. Cell Lines and Culture Method

Caco-2 and HT29 colon cancer cells were purchased from the American Type Culture Collection (ATTC), Manassas, VA, USA. The cell lines were allowed to grow in DMEM (Dulbecco’s Modified Eagle’s Medium (DMEM)) and 1X glucose (Thermo Scientific Hyclone, Logan, UT, USA). This mixture also included 10% foetal bovine serum, 1% L glutamine, and 1% antibiotic solution (penicillin and streptomycin). Cells were maintained in a humidified cell incubator at 37 °C and 5% CO_2_.

### 2.2. Cell Viability Assay to Determine Relative Proliferation in 2D Monolayer and 3D-Printed Cell Cultures

A single solution cellTiter 96^®^ nonradioactive cell proliferation MTS assay kit (Promega Corporation, Madison, WI, USA) was used to determine the viability of both 2D- and 3D-printed cultured cells, which included treated and untreated cells. In this method, 20 μL of the MTS solution was prepared. This solution was then incorporated into all of the wells in the 2D monolayer. The solution was also subjected to the 3D-printed spheroid’s culture. The set-up was then incubated at 37 °C for a period of 3 h. Following this step that lasted 4 h, both 2D- and 3D-printed cell cultures were analysed utilizing a Flurostar Omega plate reader (BMG Labtech, Aylesbury, UK). The absorbance was measured at 490 nm. The cell survival percentage in each well was estimated to compare the cellular response of Caco-2 and HT29 cells at 48 h.

### 2.3. Three-Dimensional Bioprinting of the Colitis-Like Condition Model

The INKREDIBLE bioprinter (CELLINK, Göteborg, Sweden) was employed to produce cell-laden hydrogel strands in a computer-regulated environment. A 10 µm in-plane resolution and 100 µm layer resolution were employed. The equipment contained 2 pneumatic print-points. Their extrusion capability was based on, first, the pressure exerted on the piston, second, the geometry of the extrusion nozzle’s geometry, and third, the bioink’s rheological properties. Air was pressurized and supplied with a compressor, which was set up externally. This compressor was, however, connected to an alternate outlet. The air could be pressurized to up to 250 kPa. Multiple dials could be employed to regulate the pressure transferred to the print leads. In addition, there existed printing nozzles which possessed an inside diameter of 0.6 mm. In general, the printing pressures applied were between 20 kPa and 100 kPa (Figure 1).

### 2.4. Preparation of Cell-Loaded Hydrogels

For preparing cell-loaded hydrogels, the cells were processed individually as their specific requirements demanded. The trypsinization method served to process adherent cells (Caco-2 and HT29 cells). PBS (Gibco) solution was employed to wash the culture-flasks after attaining 90% confluence, after which a solution of trypsin EDTA (Sigma-Aldrich, St. Louis, MO, USA) 0.25% (*w*/*v*) was incorporated. The area ratio employed for this addition was 1 mL/1 cm^2^ of the culture surface. After this, centrifugation was done at 500× *g* for a duration of 10 min. The resulting cells were then resuspended using the culture media. This step was followed by counting. Then, Caco-2 and HT29 were resuspended in 300 µL of the same culture medium. For uniform mixing, 3 mL of CELLINK Bioink (universal bioink) was employed for 300 µL of Caco-2 (3 × 10^6^ cells), according to the manufacturer’s instructions; the remaining 3 mL of CELLINK was treated with 300 µL of HT-29 cells (3 × 10^6^ cells). A similar method as described above was utilized. The mixing procedure resulted in obtaining 2 print-cartridges, both of which contained 3 mL of cell-loaded hydrogel: one with Caco-2 in equal proportions, and the other containing HT-29 cells. In both cartridges, the cellular density amounted to 10^6^ cells/mL.

### 2.5. Induction of Disease in 3D-Printed Caco-2 and HT29 Model

Dextran sodium sulphate (DSS) of 4% *w*/*v* concentration (wt/vol; with Dulbecco’s modified Eagle medium incorporated in 10% fetal calf serum as well as penicillin and streptomycin) was incorporated into Caco-2 and HT-29 cells. Then, it was printed to the apical and basolateral compartments of the Transwell membrane. This step was taken to induce a colitis-like scenario in our 3D-printed model [19].

### 2.6. Transepithelial Electrical Resistance (TEER) Measurements

The EVOM2 epithelial voltage test with STX2 chopstick electrodes served to measure the construct transepithelial electrical resistance (TEER) [20]. The instrument was calibrated with a 1000 Ω resistor prior to measurements, and an empty Transwell was used as a blank. Constructs were equilibrated to room temperature and switched to basal media for readings. The blank Transwell and all samples were measured three times. Samples and blanks were measured in triplicate, and then each was averaged and used in the following calculations:*R_TISSUE_* (Ω) = *R_TISSUE_* − *R_BLANK_*
*TEER_REPORTED_* = *R_TISSUE_* (Ω) × *M_AREA_* (cm^2^)
where *M_AREA_* = 0.33 cm^2^.

### 2.7. Statistical Analysis

All data were examined using GraphPad Prism 6 software using the *t*-test, one-way ANOVA, or two-way ANOVA as described in the text or figure legends. Data are presented as mean +/standard deviation. The number of biological replicates, ‘*n*’, is noted in the figure or table legends.

## 3. Results

### 3.1. The Comparative Proliferation Rate of 3D-Printed Cells Compared to 2D Monolayer Cells

We measured the proliferation rate of 3D-printed Caco-2 and HT-29 cells relative to their respective 2D cultures. Figure 2A,B show the results of the MTS assay of Caco-2 and HT-29 cells growing in 3D-printed cultures in contrast to their counterpart 2D cultures after 48 h. The result of the analysis here was that 3D-printed cells grew more slowly. This was observed in comparison to the 2D-monolayer cultures which were cultured in the initial seeding. The findings revealed that 3D-printed cell growth was 30–40% slower. There was a linear increase in the MTS assay signal with an increase in the initial cell population. However, this eventually attained a plateau when higher initial cell numbers were present. Cell overgrowth was contributed as the causative factor for this observation. To obtain a considerably intense signal when preventing this excess growth, we chose a high preliminary cell-seeding density (10,000 cells/well) for both Caco-2 and HT-29 cells for the experimental drug treatments.

### 3.2. Induction of Disease in a 3D-Printed Caco-2 and HT-29 Model

Caco-2 and HT-29 cells printed in a hydrogel using a 3D structure were treated with DSS 4% *w*/*v* for 24 h and showed a significant increase in epithelial disorganization/disfunction compared to the controls, which was assessed by the measurement of trans-epithelial electrical resistance (TEER), as depicted in Figure 3. This indicated that DSS induces a colitis-like condition in vitro [21].

### 3.3. Barrier Function Measurements of 3D-Printed Caco-2 and HT-29 Constructs

Transepithelial electrical resistance (TEER) was determined for 3D-printed Caco-2 and HT-29 cells from day 7 to day 21. Figure 4A,B show the initial increase in culture with the barrier function maintained under normal physiological conditions (dotted lines) after day 12. Furthermore, the comparative TEER values between 3D-printed Caco-2 and HT-29 cells and 2D Caco-2 and HT-29 cells were determined. Figure 4C,D show a significant reduction in TEER values in 3D-printed cells (Caco-2 and HT-29) compared to the 2D cell culture of Caco-2 and HT-29 cells, thus suggesting better epithelial barrier function.

### 3.4. Barrier Function of 3D-Printed Caco-2 and HT-29 Constructs after Drug Treatment

We examine the effect of nanoencapsulated inflammation targeting drugs (Roxadustat and caffeic acid phenethyl ester (CAPE)) on the barrier function of the 3D-printed colitis model. We can clearly observe from Figure 5A,B that 4% DSS disrupted the barrier function of 3D-printed Caco-2 and HT-29 cells. This is indicated by the highest drop in the TEER value confirming higher permeability compared to the blank, which depicts a tight epithelial barrier. Similarly, we observed that Roxadustat and caffeic acid phenethyl ester (CAPE)-fabricated nanoparticles were able to maintain the barrier function in the 3D-printed colitis model. Similarly, we observed that a combination of CAPE nanoparticles was potent enough to maintain the barrier function in 3D-printed models of cells with colitis-like conditions.

## 4. Discussion

The existing preclinical models for drug evaluation were largely limited in their capability of capturing the complex nature and physiological properties of the human gut, especially the colon, which can lead to poor predictability in drug development [22]. In this study, we attempted to incorporate primary cells into a completely human 3D-bioprinted colitis-like condition model to replicate many features of physiological colon functionality in vitro. Histological studies confirmed that a colitis-like condition was evident in 3D-printed cells.

In this investigation, it was observed that the experimental method of culturing along with the culture growth environment had a significant effect in terms of the rate of proliferation. These factors were also seen to affect the rate of proliferation of the 3D cultures as related to the 2D cultures. The type of cell lines employed decided if the proliferative intensity of a 3D culture would be fast or slow when studied together with a 2D culture. There are several cell lines, as reported in published studies, that have demonstrated a slower rate of proliferation among 3D cultures in contrast to 2D cultures [23,24,25,26,27,28]. For instance, the endometrial cancer cell lines, namely, Ishikawa (RL95-2), KLE, and EN-1078D, in a 3D reconstituted basement membrane (3D rBM) is one set of combinations among them. The other ones include the colorectal cancer cell lines Caco-2, DLD-1, HT-29, SW480, LOVO, and COLO-206F on Laminin-rich extracellular matrix (IrECM) [29]. In addition, there are also other types, such as the human mammary epithelial cell line MCF10A on a complex 3D culture system based on stromal cells, silk scaffolds, and Matrigel [30], and the human submandibular salivary gland (HSG) cell line on Matrigel [28]. In addition to the above, there is also the human embryonic kidney (HEK) 293 cell line on microspheres of cell-rat-tail collagen type I [31] that forms a part of this group.

Faster proliferative speed was seen in some of the 3D cultures, namely, the JIMT1 breast cancer cell lines that multiplied 1.86-times faster than others in the Matrigel as compared to a 2D culture [32]. In general, cells that are cultured in a 3D culture closely resembled the proliferation of tumor cells in vivo. This was because the cells that were cultured in a 2D culture were believed to be in an unnatural environment [33].

In the comparative barrier function studies, the 2D (Caco-2 and HT-29 cells) monolayer exhibited significantly higher TEER measurements when compared to their 3D-printed counterparts. This is a possible outcome of the formation of biopsy life. However, the values remained consistently above a physiological range. The physiological barrier function was successfully demonstrated through the TEER values in 3D-printed Caco-2 and HT-29 cells at day 10 and 15, respectively, and this was maintained through to day 21 of the culture. Furthermore, the measurement of the barrier in the 3D-printed construct model barrier measurement was consistent with the monolayer culture of adult intestinal epithelial cells as previously reported [34]. However, these monolayer models are typically limited to less than 11 days in-culture, can suffer from low expression of CYP similar to Caco-2 cells, and have not yet been assessed for toxicity or inflammatory response [35].

It is suggested that our data, when combined with long-term viability, indicated that the 3D-printed model has great potential to characterize other known classes of compounds that have off-target toxicity in the intestine, for example, chemotherapeutics [36]. They could help to assess and explain model chronic diseases and inflammatory pathways, for example, inflammatory bowel disease (IBD), Crohn’s disease, and colitis [36,37,38]. Future applications could raise the level of complexity by incorporating additional cell types, such as cancer cells or immune cells, and utilizing healthy and diseased donor material [39,40]. Doing so will help to better understand how each cell type contributes to the disease phenotype and better characterize candidate modulators for therapeutic intervention in disease-relevant backgrounds [40]. Regarding future direction and experimental design, we will aim to use decellularized extracellular matrix-based bioinks and explore ways to incorporate vascularization of the bioprinted tissue, which will enable us to develop a more advanced version of colon tissue with better biomimetic features. Thus, this approach will allow us to understand the biomolecular pathways of inflammation resolution via novel and conventional drugs [41,42].

In conclusion, we set out to achieve biomimicry of a fully human cells-based 3D-bioprinted colitis model with greater complexity and function compared to standard in vitro models. The 3D-printed model duplicated physiological functions and important intestinal features. It was constructed to provide a flexible platform consistent with barrier function studies.

## Figures and Tables

**Figure 1 biomimetics-08-00041-f001:**
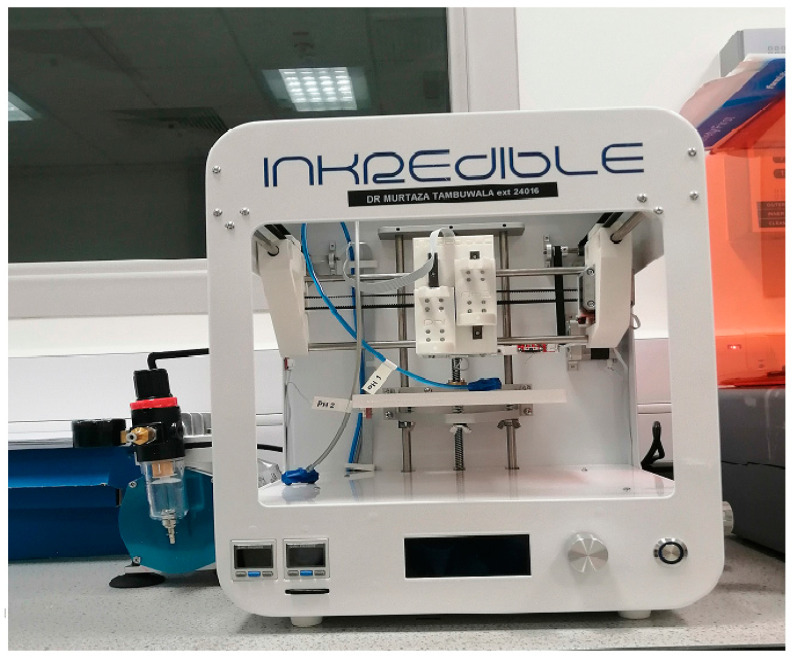
Illustration showing the pneumatic based extrusion Inkredible 3D bioprinter (dual-print head). The bioprinter employed had a UV LED curing system, which assists in bio-printing tissues or any organ model.

**Figure 2 biomimetics-08-00041-f002:**
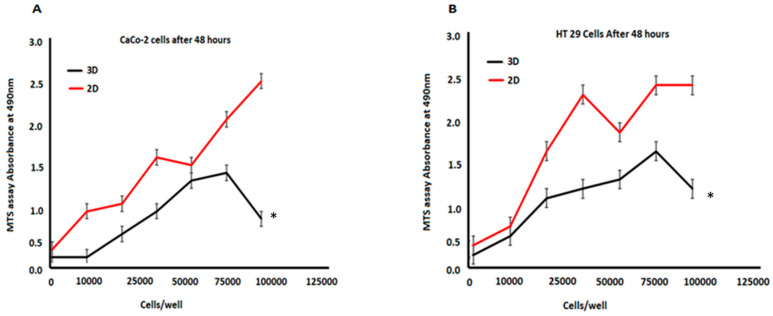
Results of the MTS assay for cell proliferation in 3D cultures and 2D cultures. (**A**) Caco-2 cells; (**B**) HT-29 cells developing in a 3D culture compared with the 2D culture (determined at 48 h). The initial cell numbers varied between 10,000 and 100,000 cells/well for Caco-2, and for HT-20, it ranged from 10,000 to 100,000 cells/well. Data are represented as mean ± SD; * *p* < 0.05 was considered significant between 3D and 2D cell models.

**Figure 3 biomimetics-08-00041-f003:**
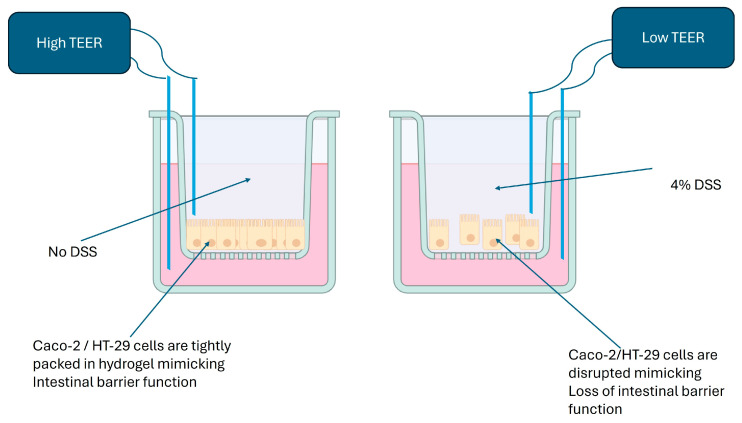
Dextran sulphate sodium (DSS) on apical side of the cells disrupts epithelial cell barrier function, which, in turn, lower the resistance in the flow of electric current as measured using a TEER reader.

**Figure 4 biomimetics-08-00041-f004:**
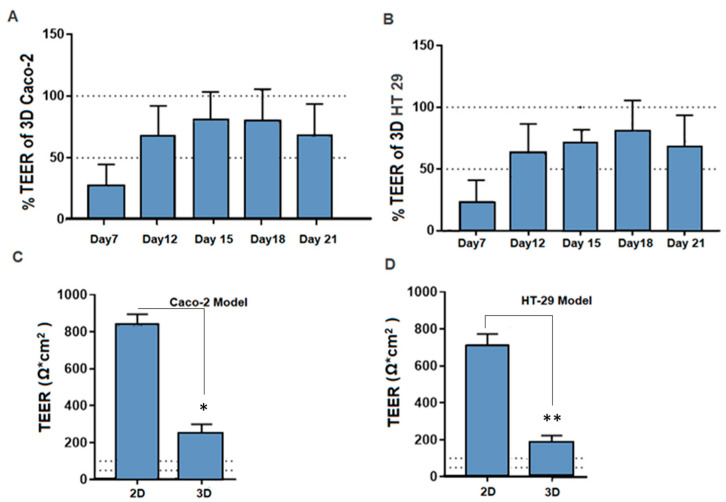
Barrier function of 3D-printed constructs. (**A**) Transepithelial electrical resistance (TEER) was determined for 3D-printed Caco-2 cells from day 7 to day 21, showing an initial increase in culture, with barrier function maintained at normal physiological conditions (dotted lines) after day 12 (*n* = 24). (**B**) The transepithelial electrical resistance (TEER) was determined for 3D-printed HT-29 cells from day 7 to day 21. (**C**) Comparison of TEER values between 3D-printed Caco-2 cells (*n* = 24) versus 2D Caco-2 (*n* = 5). (**D**) Comparative TEER values between 3D-printed HT-29 cells (*n* = 24) and 2D HT-29 (*n* = 5). Data are represented as mean ± SD; * *p* < 0.05 and ** *p* < 0.01 were considered statistically significant differences as compared to 2D and 3D cell culture models for both Caco-2 and HT-29.

**Figure 5 biomimetics-08-00041-f005:**
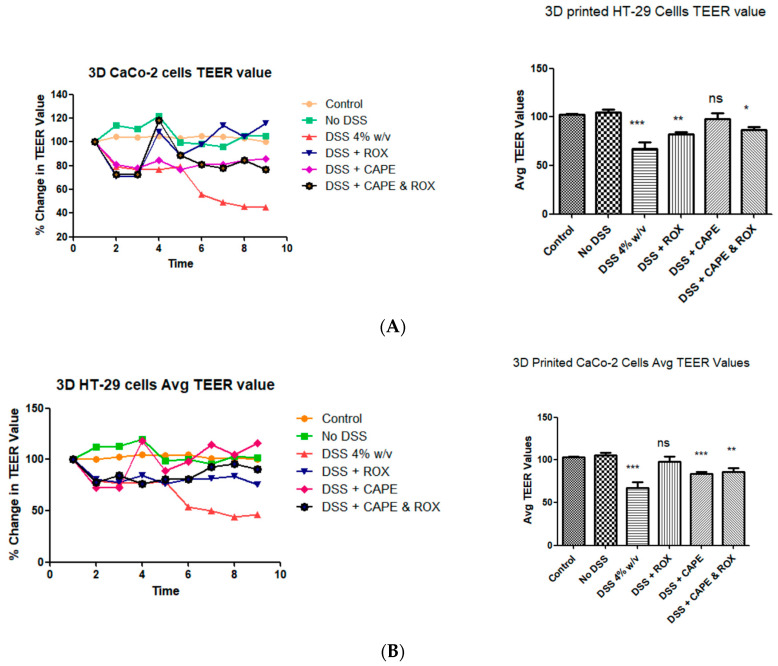
Percentage change in TEER values in 3D-printed constructs after albumin nanoencapsulated anti-inflammatory drugs. (**A**) shows the percentage change in the TEER value in the 3D-printed Caco-2 constructs, and (**B**) shows the percentage change in the TEER value in the 3D-printed HT-29 constructs. Data are represented as mean ± SD; * *p* < 0.05, ** *p* < 0.01, *** *p* < 0.001 was considered statistically significantly different compared to the control.

## Data Availability

All raw data are stored at Qassim University in a OneDrive storage folder behind a two-factor authentication wall.

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
