# Peer review of "Development of 3D-Bioprinted Colitis-Mimicking Model to Assess Epithelial Barrier Function Using Albumin Nano-Encapsulated Anti-Inflammatory Drugs"

_biomimetics, 2023, doi:10.3390/biomimetics8010041_

Round 1

Reviewer 1 Report

Almutary et al describes the development of 2D and 3D printed Caco-2 and HT29 cells used as a colitis-mimicking barrier. There are some comments to improve the quality of the manuscript as shown below.

1. In the introduction, the review of colitis-mimicking model should be included. The significance of using 2D and 3D printed models should be discussed and the novelty of this work should be expresssed.

2. The y-axis of Figure 2 should be cell viability (%).

3. According to cell proliferation results (Figure 2), what are factors resulting in the lower %proliferation of 3D printed cells compared with 2D printed cells. Although authors cited several publications to support, the discussion on rationale should be specified.

4. Line 219-220: Please check if the sentence "Figure 4C, 4D  shows significant increase in TEER values in 3D printed cells (Caco-2 and HT-29) in  comparison to the 2D cell culture of Caco-2 and HT-29 cells." Should it be 3D printed cell culture had lower TEER values?

5. Figure 5: It is interesting that DSS+CAPE+ROX-NP and DSS+CAPE-NP had shooting TEER and dropping down. Can the authors explain?

Author Response

Almutary et al describes the development of 2D and 3D printed Caco-2 and HT29 cells used as a colitis-mimicking barrier. There are some comments to improve the quality of the manuscript as shown below.

  1. In the introduction, the review of colitis-mimicking model should be included. The significance of using 2D and 3D printed models should be discussed and the novelty of this work should be expressed.

Author response: As suggested we have discussed the novelty of this work in the introduction.

  1. The y-axis of Figure 2 should be cell viability (%).
  2. According to cell proliferation results (Figure 2), what are factors resulting in the lower %proliferation of 3D printed cells compared with 2D printed cells. Although authors cited several publications to support, the discussion on rationale should be specified.

Author response: The following sentence has been added to the manuscript as per the reviewers comment

“Cell growth in 2D monolayers allows for access to the same amount of nutrients and growth factors present in the medium, leading to homogenous growth and cellular proliferation. This implies simplicity and efficiency, however, the 2D methods don’t have any control over cell shape, to control the shape of cells, adhesive islands, microwells, and micropillars have been customized.”

  1. Line 219-220: Please check if the sentence "Figure 4C, 4D  shows significant increase in TEER values in 3D printed cells (Caco-2 and HT-29) in  comparison to the 2D cell culture of Caco-2 and HT-29 cells." Should it be 3D printed cell culture had lower TEER values?

Author response: Many thanks for pointing this error on our part we have now corrected this in the revised manuscript.

  1. Figure 5: It is interesting that DSS+CAPE+ROX-NP and DSS+CAPE-NP had shooting TEER and dropping down. Can the authors explain?

 Author response: The reviewer has correctly pointed out that there was an initial drop in the groups treated with DSS+CAPE+ROX-NP and DSS+CAPE-NP – we attribute this initial drop to delayed action/ release of the drugs from the nanoparticles as compared to the free drug, also this can be attributed to the time required by the drugs to show the action on the improvement of the barrier function

Reviewer 2 Report

In this paper, the authors developed a colitis-like condition simulated model for verifying the efficacy of anti-inflammatory drugs by using 3D bioprinting. Additionally, the authors conducted a comparative experiment between the 3D structure and the 2D structure for the function of the epithelial barrier, and carried out the anti-inflammatory drug test using the manufactured 3D structure. For this manuscript to be improved, the reviewer would like to make comments as follows:

1.     The composition of the research paper was too inconvenient to read, and the meaning of comparing the groups divided by the authors is not clear. Therefore, the reviewer recommends that the authors provide a detailed description of what each group is and why the authors chose each group. In addition, although this paper is a research paper, the similarity index is too high, so the reviewer recommends that the authors should change the sentences to lower the index to less than 20%.

2.     The reviewer recommends that the authors attached an actual photo of the fabricated 3D structure. Also, the viability or function of the cell varies greatly depending on the shape or size of the 3D structure, so the reviewer recommends that the authors consider those contents or add related references.

3.     In order to accurately represent the importance of each result, the reviewer recommends that the authors add the error bars and stars indicating a significant difference between each group.

4.     In addition to the TEER value, PCR or Immunofluorescence staining using an epithelial cell specific marker seems necessary to clearly prove the integrity of the experiment. The reviewer judges that it seems unreasonable to completely explain the change in colon epithelial cell function with only the TEER value.

5.     Looking at the values of Figure 4-C and D, it is confirmed that the TEER value of the 2D group is higher than that of the 3D group. However, in the explanation of the related part, it was described that the TEER value of the 3D group was high. Therefore, the reviewer recommends that the authors should check again which result is correct and correct the contents.

6.     Because the data in Figure 2 or Figure 5 does not seem to have continuity, the authors should change the corresponding graph to a bar graph. The reviewer recommends that the authors change the TEER values to actual measured values rather than % conversion values, in Figure 4 and Figure 5 and add the result values of the 2D group to Figure 4 as well.

7.     Many other research teams have already produced 3D colon models using various hydrogels, and it has been confirmed that the models are similar to the shape of actual colons and play a similar role. There is no difference from previous tissue engineering techniques by simply fabricating the structure by stacking hydrogel. For that reason, the results of this paper were not particularly outstanding nor did it contain significant novelty. Therefore, the reviewer judged that this paper does not seem to meet the desired scope of the biomimetics.

Author Response

Reviewer 2:

In this paper, the authors developed a colitis-like condition simulated model for verifying the efficacy of anti-inflammatory drugs by using 3D bioprinting. Additionally, the authors conducted a comparative experiment between the 3D structure and the 2D structure for the function of the epithelial barrier and carried out the anti-inflammatory drug test using the manufactured 3D structure. For this manuscript to be improved, the reviewer would like to make comments as follows:

Author response: Thank you for your time and valuable supportive comments

  1. The composition of the research paper was too inconvenient to read, and the meaning of comparing the groups divided by the authors is not clear. Therefore, the reviewer recommends that the authors provide a detailed description of what each group is and why the authors chose each group. In addition, although this paper is a research paper, the similarity index is too high, so the reviewer recommends that the authors should change the sentences to lower the index to less than 20%.

Author response: the entire manuscript has been revised and the similarity have been dropped down below the requested level

  1. The reviewer recommends that the authors attached an actual photo of the fabricated 3D structure. Also, the viability or function of the cell varies greatly depending on the shape or size of the 3D structure, so the reviewer recommends that the authors consider those contents or add related references.

Author response: Unfortunately, we did not take photos of the fabricated 3D structure in the transwell plates because once we added the cell culture media it was not possible to see the printed membrane on the transwell insert wells. We have added in new related reference as suggested.

  1. In order to accurately represent the importance of each result, the reviewer recommends that the authors add the error bars and stars indicating a significant difference between each group.

Author response: we have now included this information on all graphs.

  1. In addition to the TEER value, PCR or Immunofluorescence staining using an epithelial cell specific marker seems necessary to clearly prove the integrity of the experiment. The reviewer judges that it seems unreasonable to completely explain the change in colon epithelial cell function with only the TEER value.

Author response: this is excellent suggestion by the esteemed reviewer however this was a pilot project to generate preliminary feasibility results to test the 3D printed model. In our future studies we will include PCR and Immunofluorescence staining using an epithelial cell specific marker seems necessary to clearly prove the integrity of the experiment.

  1. Looking at the values of Figure 4-C and D, it is confirmed that the TEER value of the 2D group is higher than that of the 3D group. However, in the explanation of the related part, it was described that the TEER value of the 3D group was high. Therefore, the reviewer recommends that the authors should check again which result is correct and correct the contents.

Author response: Many thanks for identifying this error on our part, we have now correct this in the revised version.

  1. Because the data in Figure 2 or Figure 5 does not seem to have continuity, the authors should change the corresponding graph to a bar graph. The reviewer recommends that the authors change the TEER values to actual measured values rather than % conversion values, in Figure 4 and Figure 5 and add the result values of the 2D group to Figure 4 as well.

Author response: we have now re made the graphs which clearly shows continuity and also made corresponding bar graphs showing average TEER values with statistics.

  1. Many other research teams have already produced 3D colon models using various hydrogels, and it has been confirmed that the models are similar to the shape of actual colons and play a similar role. There is no difference from previous tissue engineering techniques by simply fabricating the structure by stacking hydrogel. For that reason, the results of this paper were not particularly outstanding nor did it contain significant novelty. Therefore, the reviewer judged that this paper does not seem to meet the desired scope of the biomimetics.

Author response: The main novelty factor demonstrated in this manuscript is the fabrication of model for colitis using a simple 3D bioprinter, on transwell inserts. I agree there are several other manuscripts that have been published on this type of work but the fabrication method is either too complex or the 3D bioprinter used is expensive, thus the work cannot be replicated in the majority of laboratories. Our work demonstrates a simple and robust model which is relatively simple to use and economical.  

Round 2

Reviewer 2 Report

In this paper, the authors developed a colitis-like condition simulated model for verifying the efficacy of anti-inflammatory drugs by using 3D bioprinting. Additionally, the authors conducted a comparative experiment between the 3D structure and the 2D structure for the function of the epithelial barrier, and carried out the anti-inflammatory drug test using the manufactured 3D structure. For this manuscript to be improved, the reviewer would like to make comments as follows:

1.     It was very difficult to read because the revised and deleted parts and the existing contents were mixed throughout the paper. Therefore, the reviewer recommends that the authors submit a newly organized document file with only the corrections marked in a different color. In addition, because the reviewer's comments given in the previous review were not fully reflected, the reviewer recommends that the authors reflect all of these comments.

2.     Although this paper is a research paper, the similarity index is too high, so the reviewer recommends that the authors should change the sentences to lower the index to less than 20%, but it is confirmed that the attached supplementary file still does not satisfy the corresponding condition. Therefore, the reviewer recommends that the authors correct it.

3.     In order to accurately represent the importance of each result, the reviewer recommends that the authors add the error bars and stars indicating a significant difference between each group, but It is still not reflected in figure 4. Therefore, the reviewer recommends that the authors revise it.

4.     Because the data in Figure 2 or Figure 5 does not seem to have continuity, the authors should change the corresponding graph to a bar graph. The reviewer recommends that the authors change the TEER values to actual measured values rather than % conversion values, in Figure 4 and Figure 5 and add the result values of the 2D group to Figure 4 as well, but the authors did not reflect these points. Therefore, the reviewer recommends that the authors should correct it.

5.     In addition to the TEER value, PCR or Immunofluorescence staining using an epithelial cell specific marker seems necessary to clearly prove the integrity of the experiment. However, the reviewer understood the circumstances of the authors. Therefore, the reviewer recommends that the authors refer to the two papers below to design future experimental directions and think about where the manufactured platform will be applied, reflect this content in the paper, and add the two papers as reference.

(1) Kim, Byoung Soo, et al. "Decellularized extracellular matrix-based bioinks for engineering tissue-and organ-specific microenvironments." Chemical Reviews 120.19 (2020): 10608-10661.

(2) Ronaldson-Bouchard, Kacey, et al. "A multi-organ chip with matured tissue niches linked by vascular flow." Nature Biomedical Engineering 6.4 (2022): 351-371.

Author Response

In this paper, the authors developed a colitis-like condition simulated model for verifying the efficacy of anti-inflammatory drugs by using 3D bioprinting. Additionally, the authors conducted a comparative experiment between the 3D structure and the 2D structure for the function of the epithelial barrier, and carried out the anti-inflammatory drug test using the manufactured 3D structure. For this manuscript to be improved, the reviewer would like to make comments as follows:

Author response: we would like to thank the reviewer for their time and suggestions to improve our manuscript in order to make it acceptable for publication.

  1. It was very difficult to read because the revised and deleted parts and the existing contents were mixed throughout the paper. Therefore, the reviewer recommends that the authors submit a newly organized document file with only the corrections marked in a different color. In addition, because the reviewer's comments given in the previous review were not fully reflected, the reviewer recommends that the authors reflect all of these comments.

Author response: we have now removed the track changes and all correction are show in red.

  1. Although this paper is a research paper, the similarity index is too high, so the reviewer recommends that the authors should change the sentences to lower the index to less than 20%, but it is confirmed that the attached supplementary file still does not satisfy the corresponding condition. Therefore, the reviewer recommends that the authors correct it.

Author response: we have now significantly reduced similarity throughout the manuscript.

  1. In order to accurately represent the importance of each result, the reviewer recommends that the authors add the error bars and stars indicating a significant difference between each group, but It is still not reflected in figure 4. Therefore, the reviewer recommends that the authors revise it.

Author response: many thanks for this suggestion, we have now shown the statistics in form of * and P values in figure 4 C and D.

  1. Because the data in Figure 2 or Figure 5 does not seem to have continuity, the authors should change the corresponding graph to a bar graph. The reviewer recommends that the authors change the TEER values to actual measured values rather than % conversion values, in Figure 4 and Figure 5 and add the result values of the 2D group to Figure 4 as well, but the authors did not reflect these points. Therefore, the reviewer recommends that the authors should correct it.

Author response: Respected reviewer, both figures 2 and 5 have continues data we can provide the raw data as evidence, due to overlapping lines it gives an illusion of dis continuous data in figure 5 graphs, however we can assure the data is continues from hour 1. We tried to convert these graphs to bar charts but it does not convey the message correctly.

Furthermore majority of our previous research and also similar data published by other laboratory’s use % change as this gives a better representation of the change in TEER value this is due to the fact that we have multiple replicates and the starting TEER value differs for each replicate if we use the TEER value as actual measured electrical resistance it will give huge error bars, hence we convert the measurements to % change of TEER value so it gives a neat figure and will help the reader to understand the changes in a snapshot graph.

  1. In addition to the TEER value, PCR or Immunofluorescence staining using an epithelial cell specific marker seems necessary to clearly prove the integrity of the experiment. However, the reviewer understood the circumstances of the authors. Therefore, the reviewer recommends that the authors refer to the two papers below to design future experimental directions and think about where the manufactured platform will be applied, reflect this content in the paper, and add the two papers as reference.

Author response: we highly appreciate your understanding and are much obliged for your suggestion. We have now included the suggested references in future experimental section.

(1) Kim, Byoung Soo, et al. "Decellularized extracellular matrix-based bioinks for engineering tissue-and organ-specific microenvironments." Chemical Reviews 120.19 (2020): 10608-10661.

(2) Ronaldson-Bouchard, Kacey, et al. "A multi-organ chip with matured tissue niches linked by vascular flow." Nature Biomedical Engineering 6.4 (2022): 351-371.